# Aligning the “Manifesto for a European Research Network into Problematic Usage of the Internet” with the Diverse Needs of the Professional and Consumer Communities Affected by Problematic Usage of Pornography

**DOI:** 10.3390/ijerph17103462

**Published:** 2020-05-15

**Authors:** Darryl Mead, Mary Sharpe

**Affiliations:** 1The Reward Foundation, The Melting Pot, Edinburgh EH2 2PR, UK; mary@rewardfoundation.org; 2Digital Humanities, Information Studies, University College London, London WC1E 6BT, UK; 3Lucy Cavendish College, University of Cambridge, Cambridge CB3 0BU, UK

**Keywords:** problematic usage of pornography, manifesto, problematic usage of the internet, COST action network, behavioural addiction research.

## Abstract

The Manifesto for a European research network into Problematic Usage of the Internet was published in May 2018. It was written from the perspective of the COST Action Network, a programme of the European Cooperation in Science and Technology CA16207 and is expected to have significant influence on research funding priorities over the next decade. The Manifesto identified nine key research priorities to advance understanding in the field. Our analysis shows that while at the most general level it identified problematic usage of pornography (PUP) as a key research priority, it then barely mentioned it again within the body of the report. This paper uses the Manifesto’s framework to suggest research areas into the problematic usage of pornography which are of particular relevance to clinicians and other professionals working in the field who want to develop approaches to assist individuals and target groups affected by PUP. It also looks at potential research opportunities inspired by the lived-experience of users withdrawing from PUP. A large number of opportunities are identified for new work on PUP across all nine key research areas of the Manifesto.

## 1. Introduction

The publication of the Manifesto for a European research network into Problematic Usage of the Internet (PUI) [1] in May 2018 was a milestone in planning the roadmap for the behavioural addictions’ community. It provided an international focus for developing policy across different elements of the research landscape for behavioural addiction over the next decade. The Manifesto was written from the perspective of the framework of the COST Action Network, a programme of the European Cooperation in Science and Technology CA16207.

The Manifesto identified nine key research priorities to advance the understanding of PUI (Table 1).

The Manifesto identifies pornography usage as a potential PUI. We have called this activity problematic usage of pornography (PUP).

### Problematic Use of Pornography

Since 2008 pornography consumption has transitioned from the model where a market supplied physical media such as magazines and DVDs to consumers through retail networks, to a transnational, Internet-based system mainly operating on a freemium trading model [2,3,4]. In the process it has scaled from predominantly national businesses reaching an audience of millions through shops and mail-order, to a world-wide online one with perhaps a billion customers [5,6,7,8]. Today, pornography is generally accessed on smartphones and other devices. Its consumption has the potential to develop into a large-scale form of PUI [9,10,11]. This technology-driven progression has massively increased the number of people potentially exposed to PUP, while simultaneously removing barriers moderating individual levels of consumption.

Some consumers uncomfortable with the apparent impacts of their pornography use have now gathered together in large-scale, online recovery communities [12,13]. Many of these are self-help groups who try to support individuals to manage their consumption, or to end it all together. A growing community of professional therapists [14,15,16,17], coaches [18] and mental health experts [19] has developed to support these self-help groups, as well as people accessing mental and physical health services through more mainstream medical, psychological and psychiatric routes.

Academic research and public policy to support consumers, recovery communities and professionals is still at an early stage of development. Much of the motivation for writing the current paper comes from the opportunity the Manifesto offers when it says:


*“Additionally, through an interactive Dissemination Plan (including website, social media, blogs) we are reaching out to relevant stakeholders at international and national levels with an emphasis on encouraging people with the lived experience of PUI to become involved in the Action”*
[1] p. 1235.

This paper is a result of the authors heeding this call to action. First, it discusses the extent to which the topic of pornography usage is carried through in the nine key research themes. Next it considers how the Manifesto tries to address the diverse needs of the professional and consumer communities affected by PUP. It then goes on to suggest additional areas of research to bridge any identified gaps.

Our research question is “what topics should be included within future research proposals under the Manifesto to meet the diverse needs of consumers, recovery communities and professionals impacted by the problematic use of pornography”?

## 2. Materials and Methods

The current study has used the content of the Manifesto [1] as its target document. Its authors created the Manifesto both in the hope of influencing future research directions around PUI and to improve the availability of funding to support those aspirations. As far as the authors are aware, the Manifesto is the only research planning and policy document of its type currently available, which is relevant to the field of PUI.

Rather than developing a mechanism to choose to include some parts of the Manifesto and to exclude other portions, the target document is sufficiently short for it to be analysed as a single entity. Each and every paragraph referencing pornography within the Manifesto has been identified and is reproduced in this paper to provide context to allow consideration of both what has been said, and just as importantly, what has been omitted.

The Manifesto began by setting a level playing field for different problematic Internet usage behaviours. It also introduced the central focus on health impacts.


*“We use the umbrella term PUI to encompass all potentially problematic Internet related behaviours, including those relating to gaming, gambling, buying, pornography viewing, social networking, “cyber-bullying,” “cyberchondria” among others. PUI may have mental and physical health consequences”*
[1] p. 1234.

The nature of the health impacts is then developed further.


*“Disordered online behaviours, such as excessive video gaming, pornography viewing, buying, gambling, or streaming and social networks use (Ioannidis et al., 2018) have been associated with marked functional impairment including loss of productivity (or reduced scholastic achievement), and mental health sequelae including mood and anxiety disorders (Derbyshire et al., 2013; Ho et al., 2014)”*
[1] p. 1234.

The next step was to contextualise the behaviours according to their form, their place within the international medical diagnostic frameworks and as issues affecting identifiable user populations.


*“As noted, PUI envelops a wide range of activities including video gaming, pornography viewing (and other compulsive sexual behaviours), buying, gambling, web-streaming, social media use and other behaviours. Some of these behaviours may fall into an existing mental disorder in psychiatric nomenclature (e.g., gambling disorder), whereas others are likely to be formally recognized in future DSM/ICD revisions, notably Internet Gaming Disorder (Kim et al., 2016b). Different types of PUI often start in childhood or adolescence (Volpe et al., 2015), but broad age ranges can be affected (Ioannidis et al., 2018). Age and gender relate importantly to PUI behaviours, with younger people typically having problems with gaming and media streaming, males with gaming, gambling and pornography viewing and females with social media and buying (Andreassen et al., 2016)”*
[1] p. 1234.

The manifesto then moves into a process of exploring the individual research priorities according to the list above.


***“1.1. Reliable consensus-driven conceptualisations of different forms of PUI (phenomenologies, comorbidities and brain-based mechanisms)***

*The clinical aspects of some Internet-related behaviours appear phenomenologically much like addiction (e.g., gambling or viewing pornography), and demonstrate impaired control (unsuccessful attempts to reduce or cease the behaviour), preoccupation (craving), associated functional impairment (neglect of other areas of life), and persistence despite damaging effects (Billieux et al., 2015; Ioannidis et al., 2016; Kardefelt-Winther, 2017). However, it remains less clear whether, apart from gambling disorder, these other forms of PUI meet the physiological criteria relating to addiction (tolerance, withdrawal). (Fineberg et al., 2018)”*
[1] p. 1235.

This section starts to indicate at least one area of potential future research. Is viewing pornography a PUI that meets the physiological criteria for addiction? It then adds:


*“Interestingly, obsessive-compulsive personality traits are common in excessive Internet users and are associated with problematic Internet use (Chamberlain et al., 2017b), hinting that compulsive behaviours contribute to some forms of PUI. Some forms of online shopping or cybersex, on the other hand, may closely resemble ICD-10 or DSM-IV impulse control or sexual disorders (Volpe et al., 2015)”*
[1] p 1236.

However, after this fairly strong opening, placing pornography viewing into the PUI research context, we do not hear about pornography again until Research Priority 7.


***“7. Consider the impact of social factors in the development of PUI***

*For problematic online sexual behaviour (e.g., cybersex), three structural elements have been highlighted as being important contributors per the so-termed Triple A Model involving: accessibility, affordability, and anonymity (Cooper, 1998; Cooper et al., 1999), though more research is needed on this topic (Brand et al., 2016a; Wery and Billieux, 2017). Another similar proposed framework is the ACE Model (anonymity, convenience, and escape) (Young, 2008). For excessive streaming (watching videos excessively), important structural features may include the ability of given programmes to grab attention by activating a biological ‘orientating response’, mediated through techniques including the use of attention-grabbing noises, zooming/panning, and presentation of rewarding stimuli (e.g., of a sexual or thrilling nature) (Flayelle et al., 2017, 2018). Collectively, public research into the structural elements that may promote PUI in different contexts is lacking”*
[1] p. 1241.

However, this is the only mention of pornography after Section 1. So what is missing and what areas of research would be useful to the consumer, recovery and professional communities affected by problematic usage of pornography?

## 3. Results

This paper sets out to determine “what topics should be included within future research proposals under the Manifesto to meet the diverse needs of consumers, recovery communities and professionals impacted by the problematic use of pornography”?

The Manifesto tells us that PUP is a part of PUI and provides some discussion on phenomenologies, comorbidities and brain-based mechanisms. It introduces pornography in the context of how to classify it as a disorder. Some early models of pornography consumption are discussed in Research Priority 7.

Beyond these initial pointers, direct reference to the Manifesto provides little specific guidance for policy-makers considering the research environment for PUP. Unlike other behavioural disorders such as gambling and gaming, which were referred to repeatedly in the Manifesto, with pornography the Manifesto was mostly silent. It did not try linking PUP to assessment instruments, characterisation, defining clinical forms and removing obstacles to recognition and treatment. Equally it did not recommend directly relevant research on issues around genetics, personality features, interventions for prevention and/or treatments, or the field of biomarkers.

### Pornography and the Manifesto

Is the absence of more mentions of pornography viewing as a PUI in the Manifesto accidental or driven by other factors? We will not attempt to answer this. We recognise that the Manifesto’s authors covered a great deal of ground in 7000 words and some omissions were always likely. Several of the authors have published widely in the field of PUP. While still limited in comparison with research in areas such as gambling and gaming, the scale of literature investigating pornography viewing as a PUI is growing rapidly. During 2018 we read over 200 papers on pornography usage, about 90% of them published in 2017–2018.

New work which has appeared since the Manifesto was published revealed previously unquantified consumer patterns. The diagnostic category of compulsive sexual behaviour disorder (CSBD) was published in ICD-11, two months after the Manifesto appeared [20]. Over 80% of people now seeking treatment for CSBD have a pornography-use related issue, a statistic not widely known when the Manifesto was published [21]. Subsequent pornography-consumer behaviour has also been significantly influenced by entirely new factors such as the COVID-19 pandemic [22].

## 4. Discussion

So, where does an analysis of the Manifesto leave policy-makers, funders, researchers and other interested parties? Our discussion now approaches this from two separate, though linked, directions. First, we consider how PUP can be researched to meet the needs of the different stakeholder communities. We then consider the potential for future research, as seen through the lens of the nine priorities set out in the Manifesto.

### 4.1. Community-Focused Issues

Pornography accessed for free via the Internet is now used on a frequent basis by hundreds of millions of people. For most users any problems only emerge slowly over time. However, the scale of the consumer population suggests that problematic pornography usage has the potential to impact the health of millions of people across the nine key research priorities set out in the Manifesto. To make wise choices for research resource allocation, policy-makers and funders need to have sight of a research landscape that addresses each priority. What might it look like?

The following is a pragmatic assessment based on a consideration of the existing literature, mediated by the constraints set within the Manifesto. This assessment is supported by the author’s wide-ranging dialogues with professional, recovery and consumer communities. Since 2016 we have had face-to-face discussions on pornography consumption with over 9000 people in the United Kingdom, the Republic of Ireland, France, Germany, Croatia, Hungary, the Ukraine, Turkey, Japan, Australia and the United States [23,24].

#### 4.1.1. Professional Communities Affected by Problematic Usage of Pornography

What else do therapists, medical practitioners, counsellors and sex educators want to know about the mental and physical health implications of problematic pornography usage? What are the issues that their clients, the problematic pornography users, want to have investigated? It is unhelpful that the diagnostic manuals do not mention the word “pornography” when over 80% of people seeking treatment for compulsive sexual behaviour have a pornography-related issue [21]. Further, the nature of the condition for many meets the criteria for addictive disorders and should be clearly categorised as such to enable healthcare professionals to respond with appropriate treatments. Both groups have an interest in distinguishing between classifying pornography usage as a potentially addictive disorder versus it being considered as an impulse control disorder. At any rate, there is clear desire in both lay people and professionals to have the word “pornography” appear in any given classification of a future edition of the International Classification of Diseases [20] and the Diagnostic and Statistical Manual of Mental Health Disorders [25].

It is apparent from our experience in meeting researchers from all parts of the world [23,24] that the therapists’ own knowledge, attitudes and prejudices are very important issues in terms of delivering access to appropriate treatment services. While the Internet brings PUP to all parts of the world, different cultures have different taboos and even blind spots about what is PUP, what is unacceptable behaviour and what sorts of treatment might be considered appropriate. The legal systems of different countries are also relevant, and there are no universal standards for most sexual behaviours.

For the last four years the authors have been teaching healthcare professionals about Internet pornography and behavioural addiction. These audiences have consistently expressed a desire for the health and social implications arising from pornography usage to be taught as a component of general medical education. Clinicians have also expressed interest in integrating the management of compulsive sexual behaviour disorder into the regular functioning of national healthcare systems.

#### 4.1.2. Issues within the Pornography Recovery Communities

It would be desirable for the Manifesto to address the concerns of the online pornography recovery communities and members of 12-step programmes such as Sex Addicts Anonymous [18,19,26]. So far, we know of no quantitative research that has investigated elimination of digital porn use to reverse sexual dysfunction and a variety of mental health disorders reported by those who quit [27].

The focus within recovery communities begins with recognition/identification/diagnosis of problematic pornography usage. The first question is “do I have a problem?” If they do have a problem, this quickly becomes “how do I stop?” with a focus on tools, techniques and support. It then becomes “how do I maintain my desired level of sexual sobriety?” Here we need more longitudinal research on the mechanisms and techniques for overall programmes of quitting pornography usage, on abstaining from pornography viewing, and the issue of managing masturbation. Unlike most PUIs such as gambling, shopping or gaming, pornography viewing is linked to very deep biological sexual and reproductive drives, with the option of the interactive reinforcement by masturbation. The pornography, masturbation and orgasm (PMO) cycle reinforces the viewing behaviour, making simple abstinence strategies difficult to maintain for many former users. When you stop gaming you can get rid of your game console or you can have yourself banned from gambling venues, but you cannot stop being a sexual being.

There are some specific characteristics of problematic pornography viewing which merit more research, particularly issues around triggering and flatlining. A deeper understanding of PUP triggering factors would be useful to help individuals and clinicians build better strategies and models for avoiding problematic usage. Flatlining is the term for the severe and sustained, short to medium-term loss of libido some users report upon quitting a cycle of problematic pornography viewing. While widely reported in community forums such as NoFap [12] and RebootNation [13], it is not well-covered by the academic literature [28].

The medical establishment has an ongoing interest in the development of new approaches to support quitting pornography viewing by identifying effective drug treatments. Naltrexone is a medication that is used often in drug and alcohol addiction. It blocks the effects of drugs known as opiates. It competes with these drugs for opioid receptors in the brain. Naltrexone can reduce a patient’s desire for drinking thereby helping them remain abstinent. Three papers have looked at the effect of naltrexone on problematic pornography use [29,30,31]. Given the similarity of brain changes in substance abuse disorders and behavioural addictions, this could prove a fruitful area for further research. Very recently transcranial magnetic stimulation has been suggested as a novel treatment [32]. Investment in research to try to demonstrate the value of such approaches to clinical populations at scale would be very helpful.

Therapists operating within the context of organisations such as the Association for the Treatment of Sexual Addiction and Compulsivity (ATSAC) in the United Kingdom [16] and the Society for the Advancement of Sexual Health (SASH) in the United States [15] would benefit from more research into behavioural modification techniques to help their clients stop Internet-based behaviours. This might include the creation of apps and other Internet tools. Very recently metacognitive therapy has claimed to be three-times as effective as traditional cognitive behavioural therapy. It deals successfully with ‘desire thinking’ and rumination, a behaviour that often predicts relapse. Extending new work on this to clinical populations of recovering pornography users would be helpful [33,34].

#### 4.1.3. Consumer Communities Affected by Problematic Usage of Pornography

Pornography viewing is now a widespread behaviour in most countries where there is good quality access to the Internet [8]. To have a protective effect on health and wellbeing, consumer-facing academic research on PUP needs to have impact in the wider community. This is a science communication issue. Little published research has looked at the ecology of spreading this type of message. There is scope for developing a very diverse array of effective communication strategies and toolkits. More research is needed on what would help adults to avoid situations and behaviours that might lead to PUP.

There is more active research focused on schools and adolescents. The Manifesto could support the sort of educational work The Reward Foundation does in schools to help equip young people with the knowledge about the reward system of the brain and the skills they need to avoid not just PUP, but PUI in all its forms [35]. Other educational programmes are available from an array of non-governmental organisations such as Culture Reframed, eChildhood, and the Naked Truth Project. Common to all programmes is a lack of longitudinal studies evaluating their effectiveness over time, across cultures and among varying age groups.

We also need to recognise that PUP is not just a first-world issue. The Reward Foundation website operates in English, but uses the machine translations system GTranslate to make all pages available in over 100 languages. As a result, a significant proportion of our traffic is now in African languages such as Hausa and Somali. These people have a need for information about PUP, but it is not readily available from within their own language communities.

World-wide data from Pornhub shows that heaviest usage of pornography is by people in the 18 to 34 years age brackets. It is strongly tilted towards male consumers, but supply-side drivers are slowly pushing consumption towards gender parity [5,6,7,8].

Research over the past decade has shown that pornography viewing correlates with higher levels of sexual assaults and dating violence, pathways into domestic violence and reduced levels of bystander intervention [36,37,38,39]. The next generation of research should help move us from noting these as correlations to investigating questions of causation (or not) in these fields.

### 4.2. Future Research Seen Through the Lens of the Nine Priorities

The categories in the following discussion are directly based on the Manifesto’s nine research priorities, simply substituting ‘problematic usage of pornography’ for the wider PUI category. They map out some of the ways the Manifesto could respond to the needs of the professional and consumer communities.

Research area 1: Reliable consensus-driven categorization of Problematic Usage of Pornography (defining main phenotypes and specifiers, related comorbidity and brain-based mechanisms) [1].

Separate and interlocking strands of research would be useful in the two main fields analysing sexual behaviour; compulsive sexual behaviour towards people and the usage of pornography. In particular, the overlap between them needs more work. There would also be considerable value in research on the overlap of pornography viewing with other PUI’s involving sexual behaviour-related activity, especially dating apps and social media.

Research area 2: Age- and culture-appropriate assessment instruments to screen, diagnose and measure the severity of different forms of Problematic Usage of Pornography [1].

The number of screening, diagnostic and measuring tools for pornography usage keeps expanding. There would be considerable value in testing and calibrating the best of the existing crop of tools, such as the Brief Pornography Screener and Problematic Pornography Use Scale, across genders, age groups and societies [14,40]. Extending field trials would materially improve the ease with which these tools can then be rolled out for widespread adoption by clinicians and other people who are in the first line of contact by individuals concerned about PUP. Even among the experts the Brief Pornography Screener is not universally known [41]. Separately, major international reviews of erectile dysfunction have not referred in any way to the behavioural addiction field for data [42].

Tolerance and escalation are classic behavioural symptoms of the journey towards an addictive disorder. Each contributes to some consumers developing problematic usage. More research would be useful in relating the roles of tolerance and escalation in the PUP of viewers who escalate to viewing child abuse material. This seems to have become a wide and growing aspect of cybercrime around the world [43,44]. More knowledge of effective crime-fighting strategies could lead to better policy-making on prevention and to better risk-based advice for the courts and judges around sentencing and disposal. Standardised and tested risk assessment tools would be particularly useful.

Problematic usage of pornography seems to be creating a new separate sub-population of consumers who choose to view child abuse material without having any particular sexual interest in children in real life. They appear to be a separate clinical population to individuals with a paedophilic sexual orientation.

Another area where screening tools are woefully lacking is in the categorisation of the use of pornography by people on the autistic spectrum. Whereas about 1% of the population have autism spectrum disorders, crime statistics suggest that a disproportionately large number of the people (circa 30%) who escalate to viewing child abuse material come from specific vulnerable communities such those with as autism spectrum disorders, ADHD and those with learning difficulties [35]. These vulnerable populations are often considered ‘asexual’, and not requiring any sex education. This is wrong as many use the Internet as a substitute for relationships that they find so difficult to negotiate in real life. People with ASD are rarely represented in other crime categories. Their emotional immaturity and inability to interpret social rules can result in an attraction to younger people online who match their emotional age better. More research needs to be done on this group to improve prevention of access to child abuse material and to consider their special needs in terms of disposal following a conviction. For example, they do not function well in groups, which are the typical types of setting for sex offender rehabilitation [45].

Research Area 3: Characterise the impacts of problematic Usage of Pornography on health and quality of life [1].

#### 4.2.1. Is Problematic Pornography Usage a Primary Disorder?

This question can be explored by research into the pornography, masturbation and orgasm cycle. In turn, this line of work could lead to consideration of the extent to which PMO may be driving depression, social isolation and other mental health issues in the wider population. Is it a bi-directional process – to what extent are these mental health issues powering the PMO cycle? Work in this space to test causation was recommended as far back as 2016 [27], but has yet to gain funding.

Bi-directional work would also be helpful for studies exploring health links or problematic pornography usage and the Big-5 personality factors. First, factors such as narcissism are regularly shown as correlates of heavy pornography usage [46], but few studies look at what happens to personality factors after people end problematic pornography use. Do they remain constant or is there an individual and/or population tendency for them to revert to what they were before the individual began engaging in PUP [27]. Other recent studies have highlighted the need for longitudinal studies in this area [47].

Similarly, conference reports [23,24] and our informal discussions with both individual problematic pornography users in the community [26] and among sex therapists [15,16], suggest that there are co-morbidities with other addictions, both to substances and behaviours. Developing new and/or improved strategies for treating complex webs of problematic usage are always of value.

More literature would be helpful on the impact of problematic pornography use at different points in the human life-cycle. Most research published to date has focused on teens or university students, though around half of all pornography viewers are older [8]. This makes case studies or more specific work on older people very rare [48,49]. It would also allow us to look at the potentially very different trajectories of younger populations who began with access to unlimited, free streaming videos in or around 2008, and older users who may have transitioned through several successive modes of access to pornography such as magazines and DVDs, to Internet use.

#### 4.2.2. Individual, Relationship and Community Impacts

The many roles of pornography in forming, sustaining and ending couple relationships would benefit from further research. The majority of research to date has been on users, and to a lesser extent their sexual partners. There would be value in unpacking the factors relating to pornography consumption as they impact on various aspects of an individual’s health and quality of life in the context of a dyad [18,19,28]. This focus on couple relationships naturally tends to exclude work on the lack of relationships among young heavy users, around 50% of whom are virgins and not in relationships [12]. The other area needing more work is on the way pornography creates expectations of “how partnered sex will be” [35,46]. The literature of wider family and community impacts of pornography use across the human lifespan is also limited [48].

Additional research into the impact of accessibility would be valuable, as problematic pornography usage would seem to be a case where the PUI strongly links to private viewing. This is especially true for mobile platforms which can be used alone or with sexual partners. Smartphones are also now used widely for technology-enabled flirting including “sexting” with diverse relationship, social and potentially legal implications.

How does pornography consumption relate to the health and quality of life in a couple or a family? The authors frequently train social workers and criminal justice professionals who report that consumption of pornography, particularly violent pornography, is a consistent negative factor in the out-of-control domestic situations they encounter in their work. Unlike mainstream television where intimate partner violence in programmes tends to be simulated, most violence depicted in pornography is real [37,50]. It is not simulated and may offer viewers ethically challenging issues while they are viewing. It may also lead to post traumatic stress disorder and related symptoms after they stop. Once seen, real violence and intimate partner violence shown in pornography cannot be unseen. Where the real rape of minors is being streamed as entertainment by major commercial pornography sites, the legal and ethical challenges can be even more complex [51].

To what extent does acting out violent, coercive or controlling pornography scripts with real partners drive domestic violence and a growing appetite for dangerous practices such as sexual strangulation [52]? These questions are key in many different cross-disciplinary approaches to understanding PUI and PUP.

Research Area 4: Define the Clinical Courses of Different Forms of Problematic Usage of Pornography [1].

The Manifesto noted that:


*“Remarkably little prospective research has been conducted on the courses of different forms of PUI and we remain relatively ignorant of key factors affecting long-term out-comes. Such data are of crucial importance in understanding aetiology, planning treatment and improving prognostication. For example, for some individuals PUI may represent a temporary phenomenon and spontaneously resolve (e.g., in some young people as brain systems mature), whereas for others PUI may become chronic.”*
[1] p. 1238.

These observations apply very strongly to PUP. There are many areas where we have little data.

In simple terms, PUP can lead to the individual failing to stop their use when faced with consequences, following the model set out in the diagnosis for compulsive sexual behaviour disorder [20]. However, ICD-11 is careful not to set out the details of the specific sexual behaviours that are causing the distress. Equally it is vague about the progression of the disorder from the perspectives of screening, assessment and diagnosis. This gives considerable scope to clearly define the clinical courses of each and every form of PUP.

We suggest that there are two fundamental classes of PUP, and each needs more research. The first is where the consumption of pornography leads to problems directly attributable to the consumption process. The user cannot stop their use and it is leading to negative consequences. This is applicable to CSBD as an impulse control disorder [20] and would also apply if a future version of the International Classification of Diseases gave it an entry in the addictive disorders section.

We propose that the second category is where the development of a PUP comes from the ideas it introduces into a viewer’s mind and life. These can range across a whole spectrum from requiring to replay sexual scripts from pornography in your head to get aroused during partnered sex, to learning to ignore the need for consent in sexual activities, to promoting rape myths, the diminution of bystander intervention and the promotion of practicing dangerous activities such as autoerotic asphyxiation. Other problematic uses can encompass escalation to criminal activities around viewing or creating images of child sexual abuse. It can also lead individuals to engage in sextortion, ‘capping’, and other illegal activities towards minors [35]. Other new populations are the technology-enabled voyeurs whose illegal activities are also directed at adults, encompassing upskirting, hidden cameras, revenge porn and sharing sexual material without permission. Pornography has also appeared that was secretly recorded on baby monitors lacking password protection.

Pornography-induced erectile dysfunction (PIED) [28,53] has been identified as a key factor that causes male sufferers of PUP to seek treatment [54], but this area still lacks clear before and after brain-imaging studies [27]. PIED has also been linked to escalation to child abuse material. As desensitisation, tolerance and hypofrontality develop, a problematic pornography user needs more shocking material to feel any arousal.

Therapists need to know how long it can take for a PUP to develop and what external or internal factors can act as triggers. Reports from the recovery communities talk about the triggering influence of everything from life events and rites of passage, to social networks and the manipulation of viewer’s taste by the algorithms used by the commercial pornography suppliers.

The withdrawal process from PUP, both physical and mental, seems generally similar to that of other PUIs. The only unique characteristic of PUP identified so far is ‘flatlining’. The literature does not adequately characterise the features of flatlining as a symptom across the population of users withdrawing from PUP. This means that when it happens, problematic users can relapse to try to feel aroused again and overcome their feeling that their sexuality might be permanently ‘broken’.

More studies of co-morbidity, both between PUP and other behavioural or substance addictions, and with other compulsive sexual behaviours, would also be welcome.

There are other elements research could reveal about the general characteristics of PUP. What are the characteristics of the incubation periods of PUP at both the individual and population levels? Do they have characteristics driven by gender or other factors? This in turn leads to the need for wider studies looking at gender, lifestyle and sexuality-based approaches to tease apart key elements in the PUP process. What commonalties and differences exist across the LGBTQI++ spectrum? In parallel, is PUP in the chemsex community driving addiction, or is its influence bi-directional [55]?

Existing brain-imaging studies [56,57] hint that both the quantity of pornography viewed and the period the viewing is spread across can have impacts on the development of PUP. However, the current crop of brain studies on pornography are restricted to people in their 20s and 30s and they also draw on very narrow demographic and cultural samples. In a world where an average age for boys’ first exposure is 12 years or younger, brain development issues need investigation [2].

More women-focused studies would be beneficial. How is PUP the same for women as for men, and how is it different? Are the genders the same in the etiology of compulsive use? To what extent is sex toy used [58] a component of women’s PUP? Some research has suggested that pornography usage by women has influenced their behaviour in couple relationships [59,60]. Are there differences according to ages and points in the lifecycle? We are unaware of any PUP studies in women controlling for stages across the menopause as a variable.

Age and stage of life should also be addressed from other perspectives. Virgins make up perhaps 50% of the recovery community [12]. What are the longer-term implications of early PUP across their lives as sexual beings? Separately, how do the challenges faced by older addicts, who used a lot of pornography before the Internet, compare to those of purely Internet-based consumers?

What are the impacts of PUP on relationship formation, sustainability and sexual health? A study of Korean men in stable relationships suggested they had developed a preference for viewing violent pornography as an alternative to real sex with their dyadic partner [61]. This also raises the issue of the nature of the content of the pornography being viewed. In line with the self-reports from the recovery community, a Swedish and Italian study of university students demonstrated that unusual sexual content creates more excitement. It also concluded that lots of the acts considered “unusual” by the researchers in 2015, already had become mainstream to many viewers and did not seem “unusual” at all [62].

Research Area 5: Reduce Obstacles to Timely Recognition and Interventions [1].

Obstacles around the timely recognition of PUP and in delivering interventions exist both within the user population and among professionals. The interests of the groups overlap where there is a desire to develop, test and deploy more public health intervention strategies to draw the attention of pornography users across society to indicators of problematic usage.

At the widest level there is value in developing strategies to have the issue of PUP accepted by national healthcare systems. It can now be diagnosed as compulsive sexual behaviour disorder, coded 6C72 under the parent category of impulse control disorders [20]. Some clinicians are using this categorisation, but there will be considerable lags before it is adopted by nations world-wide. In some countries such as the United States, the need for clinicians to have a code for CSBD to allow patients to claim on medical insurance is paramount.

A separate issue revolves around making effective training for professionals more widely available. There is a need for the right balance of availability of practitioners (numbers, skills, geographical spread) to the scale of needs of people with issues.

Training can be built into continuing education programmes for professionals, as it is in the United States through the Society for the Advancement of Sexual Health and the International Institute for Trauma and Addiction [15,17]. In the UK this is done through the Association for the Treatment of Sexual Addiction and Compulsivity and the College of Sexual and Relationship Therapists [11,16]. However, CSBD and PUP are very new ideas and they have not become part of general medical training in most tertiary institutions.

Research Area 6: Clarify the Possible Role of Genetics and Personality Features in Different Forms of Problematic Usage of Pornography [1].

This area of PUP research is still very immature. There would be value in developing a broad literature looking at the relationship between genetic factors and the natural history of problematic pornography usage. Are particular individuals or populations especially vulnerable? Key dimensions might include gender, sexual orientation and exploring health links to Big-5 personality factors.

Proof of concept was demonstrated by a 2017 paper [63] examining links between gene methylation and hypersexual disorder. The study found that the epigenetic state in the CRH gene may contribute to explaining the biological mechanisms of hypersexual disorder. Thus, some genes appear to play a role in hypersexual disorder.

Within the recovery community there is a strong desire to find answers to the ‘why me’ question. What is the relationship between predisposition to PUP versus acquiring it through consumer behaviour in a changed technological environment? Epigenetic factors need to be further researched. Informal evidence from the online recovery communities indicates that certain character traits or personality types thought to be fixed actually change when regular use of a supernormal stimulus such as hardcore pornography ceases. For example, in some users out of control hypersexual or narcissistic or sensation seeking or aggressive behavioural traits remit or disappear when the stressor is removed.

Research Area 7: Consider the Impact of Social Factors in the Development of Problematic Usage of Pornography [1].

Several separate, but overlapping strands of research around social factors would be useful. Five are highlighted here as requiring more investigation: the technologies of the attention economy; women’s changing role as consumers; environmental impacts; social acceptability; and the potential for early exposure to pornography to be considered as an Adverse Childhood Experience.

First, the Manifesto in Research area 7, in Section 2 above, identified


*“For excessive streaming (watching videos excessively), important structural features may include the ability of given programmes to grab attention by activating a biological ‘orientating response’, mediated through techniques including the use of attention-grabbing noises, zooming/panning, and presentation of rewarding stimuli (e.g., of a sexual or thrilling nature)”*
[1] p. 1241.

This neatly summarises the core behaviour of people engaged in PUP, where watching is generally accompanied by the viewer masturbating. This is absolutely an area where public research is required into the structural elements of commercial pornography sites as a part of the attention economy. Commercial sites seem to have adopted the structural concepts pioneered by B.J. Fogg at Stanford University and the behavioural design work of Nir Eyal [64,65]. This allows them to build websites and apps that change users’ thoughts and behaviour without their knowing. In particular, the pornographers target the unconscious mind via dopamine pathways in the reward centre to stimulate cravings that keep users coming back for more, resulting in PUP for increasing numbers of consumers. Artificial intelligence is being deployed by the commercial pornography suppliers to learn about user’s sexuality to deliver the most engaging experience possible. This has implications for privacy. As some research suggests that pornography viewing can change a person’s sexual tastes, this is a very slippery area. Do we really want a big-porn machine algorithm to reshape our sexual templates [66]?

Another area ripe for research is the issue of intentional manipulation of consumption patterns by commercial interests in the pornography industry. In particular, since 2016 there has been a sustained increase of usage by women as a world-wide phenomenon [5,6,7,8,67]. To what extent has this resulted from deliberate female-focused promotional strategies of the commercial pornography suppliers and what are its implications for dyadic relationships, family structures and society in general?

Pornography sites get heavy, repeat traffic from users. The largest supplier, Pornhub, streamed 115 billion videos in 2019 to a world-wide audience of perhaps 1 billion individual viewers [8]. It would be extremely helpful to have reliable calculations for the world-wide use of pornography. The industry is now so large that it has been calculated to consume 27% of the energy used by the Internet and is currently responsible for 0.2% of all greenhouse gas emissions world-wide, roughly equivalent to the domestic energy use of every household in France [68]. The environmental impact is large enough for researchers to calculate a quantifiable contribution to rising sea levels.

Across cultures and within cultures, the phenomenon of “normalisation” of pornography consumption as a socially acceptable activity has grown rapidly. Since the arrival of the iPhone in June 2007, supported by high-speed network services, the smartphone supplying “free” commercial pornography has removed barriers and increased temptation through its Triple-A model of effectively unlimited access, affordability and anonymity [69]. The speed of this adoption has outstripped the capacity of researchers to place it in context and to include it in theoretical models. It is clear that the arrival of smartphones, Wi-Fi and 4G networks leads to most societies adopting pornography as a more acceptable form of recreation, especially among the young. The trajectory of this change can be unique to each country [2].

Many stories circulate within the recovery community suggesting that childhood exposure to pornography in general, and Internet pornography in particular, should be considered an adverse childhood experience (ACE). However, at present this cannot happen. The list of definitions of ACEs pre-dates the arrival of problematic usage of pornography from the Internet. It would mean revisiting the research field of defining, testing and verifying the validity of PUP as an ACE. This would be a very worthwhile project.

Research Area 8: Generate and Validate Effective Interventions, both to prevent Problematic Usage of Pornography, and to Treat its Various Forms once Established [1].

#### 4.2.3. Supply-Side Preventative Interventions

These tend to revolve around whole-population approaches to changing the ease of access to Internet pornography. After three false starts, will age verification legislation in the UK prove to be a successful intervention? How and where might it be tried in other countries? How will it be validated and will it achieve the intended outcomes?

There is an established and diverse commercial ecosystem of blocking technologies now available for individuals and groups to prevent access to Internet pornography. Their characteristics tend to be specific to their local geographies. They can operate at the individual level, usually by subscription. Many Internet Service Providers (ISPs) offer filtering on either an opt-in or opt-out basis. Academic analysis of the effectiveness of these measures would be helpful.

PUP research could also look to public health models used for other areas of addiction, particularly online gambling, where the business models and delivery mechanisms are often quite similar. Policy research into new models of control, monitoring and product delivery limitation involving the government, commercial and voluntary sectors could be developed. This remains minimally charted territory.

A recent development in pornography delivery (and sometimes production) is the increasing role of social media as a point of access. While games delivery platforms are now a significant source of pornography for some consumers, we have also encountered adolescents getting around parental controls via the Internet of Things. A poorly protected smart refrigerator could get a grounded teen online.

#### 4.2.4. Supply-Side Treatment Interventions

Individual or community approaches to treatment from counsellors, schools, churches or other local units can be successful, especially when linked to demand-side reduction measures. Accountability partners are popular in some communities [26]. Again, more external validation of success, or not, of different approaches would be valuable.

#### 4.2.5. Demand-Side Preventative Interventions

Are all pornography consumers at risk of developing PUP? The simple answer is probably not, but we do not know for certain. If the general characteristics of PUP are broadly similar to other better studied behaviours with a propensity to lead to behavioural addiction, the answer is ‘no’. A proportion of users will be likely to move first into problematic use and then addictive use. Some studies suggest that all pornography consumption generates brain changes, even at quite low durations [56,57]. So far, the World Health Organization only considers the research base is strong enough to support the diagnosis of compulsive sexual behaviour disorder under the parent category of impulse control [20]. At the same time, few pornography consumers have been exposing themselves to large volumes of online erotica for longer than two decades, so we have only explored it as a factor for, at most, half of the duration of a human beings’ sexual life-cycle. We do know that the more pornography some people consume, the more likely they are to develop compulsive behaviours as a result. It is the sort of issue that could respond well to the development of a risk-managed framework. Such a framework could balance benefits of pornography consumption at the individual and society levels with the inevitable side-effects from some individuals being unhappy with the consequences arising from their use.

Where consumers do develop concerns, many practitioners believe that the solution to PUP is rooted in consumer education. As a society we can choose to try to reduce the strength of demand for access to pornography. Some individuals who see pornography as a valuable part of a sex positive culture would be uncomfortable about this, but it is up to each society to consider what restrictions may or may not be appropriate if some individuals may be harmed by excessive consumption.

More systematic assessments of the success of educational strategies would show how effective they can be as risk-reduction strategies. What role will learning about the addictive potential of PUP play in reducing demand? Does learning about porn-induced erectile dysfunction reduce consumption? Thousands of anonymous self-reports on the recovery websites show that the availability of knowledge of the effects of Internet pornography on the brain was sufficient to encourage users to experiment with quitting porn [12,13]. When they did, a range of mental and physical health problems remitted or cleared up. Only when heavy users quit porn did they realise that those conditions were related to their porn use. Formal research into this phenomenon would be valuable.

A separate investigation into the issues of prevention versus cure would be interesting. We are suggesting research to compare the benefits of never beginning pornography usage versus treating the problem when it has been ‘cured’. At a population level, are people who quit ultimately as mentally and physically healthy as those who never engaged in PUP in the first place? What methodologies might help unpack this question?

#### 4.2.6. Demand-Side Treatment Interventions

The development of possible drug treatments is still at an early stage. In theory adjusting brain chemistry with substances could reduce libido and cravings for pornography. The most common approach is to look at substances that are already approved for clinical use and to begin the process of taking them through stages from case reports and on to small and then large-scale clinical trials. What drugs might discourage PUP and what pathways should be targeted? A major review was published by Sniewski and colleagues in 2018 [70]. Early work focused on possible similarities to obsessive compulsive disorder and suggested trying drugs known as selective serotonin re-uptake inhibitors (SSRIs). Recent case reports have been published for Paroxetine [71], Nalmefene [49] and Aripiprazole [72], with mixed results.

Research Area 9: Identify Biomarkers, Including Digital Markers, to Improve Early Detection and Intervention [1].

This is a new area, but it may have real potential to help improve health at the individual and population levels. As far as we are aware, biomarkers are not currently in use for detecting or diagnosing PUP. Would there be value in their introduction?

Research might be able to separate out the underlying tendency of some individuals or groups to be susceptible to problematic usage. This could be contrasted to the potential for Internet pornography to lead to problematic usage as a natural characteristic in and of itself. The issue here is “are you the problem?” versus “is pornography the problem”. Co-morbidity studies involving other PUIs would also be helpful—many users find themselves trying to unhook from gaming or recreational drugs and pornography at the same time.

Biomarkers are a high-tech intervention. In face-to-face situations, such as a doctor’s surgery or a counsellor’s office, are they a better solution than simply asking a person about their level of pornography consumption? The brief pornography screener with five questions on a single A4 sheet of paper can be administered in three minutes and seems to be both sensitive and reliable [14].

A different line of research would be to look at the potential for widescale interventions using Internet data. It might even extend to artificial intelligence algorithms at commercial pornography suppliers or the ISP level. It would mean monitoring people for signs of problematic usage and then invoking a warning or even a cut-off protocol. This has privacy and data management implications, but the reality seems to be that monitoring rather like this is already at the core of the business model of major commercial pornography suppliers. Could governments consider introducing national level intervention programmes? How would they operate and what would be the success criteria?

At the same time, the pornography supply companies are very reluctant to share their data. In this they follow the lead of other industries supplying addictive substances like tobacco and alcohol. The commercial pornography suppliers are quite vulnerable to a ‘smoking gun’ which showed that were trying to induce addictive behaviour as a part of their business model.

## 5. Conclusions

When you consider the conclusions in the Manifesto [1] in the light of the current paper, there is nothing recommended for research into PUI that does not equally apply to the sub-discipline of PUP.

To come back to our research question, “what topics should be included within future research proposals under the Manifesto to meet the diverse needs of consumers, recovery communities and professionals impacted by the problematic use of pornography”?

The comments in this section have been grouped to answer these questions individually, but there is, and should be, overlap between them, so we begin by looking at general questions about the field of research. The management and treatment of PUP requires the right approaches to meet the specific needs of everybody in this mental and physical health ecosystem.

### 5.1. General Research

Given the massive scale of use of pornography and its potentially negative outcomes for some users, it is important for the European Union’s Problematic Usage of the Internet (EU-PUI) Network to have a visible and trusted champion for PUP, just as it should for other behaviours. Linked to this, Europe should hold a planning conference under the umbrella of COST Action to design its response. The opportunity for PUP data to be made accessible to researchers through shared multinational databases is extremely attractive.

To assist the World Health Organization in improving the classification of compulsive sexual behaviour disorder in future to include the word pornography, research is required to separate out the natural history of PUP from compulsive sexual behaviours focused on people.

Experimental work to investigate causation through longitudinal studies would provide a better strategic foundation for PUP. This would be stronger if it included large-scale brain imaging studies to gather evidence before and after people have quit or have been given treatment. It is essential to develop a wider evidence-base covering all ages and stages of the human life-cycle for people engaging in PUP. There should be much more research focus on women as they are now the main sector of growth for PUP. Equally our data should embrace sexual diversity and investigate the usage among LGBTQI++ communities where levels of mental health disorders and chemical dependencies are higher than in the mainstream heterosexual population. Could PUP be a contributing factor?

### 5.2. Research for the Professional and Therapist Community

The 2020s is the decade when training of health and social care professionals should begin to incorporate a deep and broad understanding of all PUIs as a matter of course. This will be a communication and public relations programme. It should be recognised that within the field of PUP we are starting from a low-base with modest levels of training or resources available for people working as professionals and therapists. New research should be leading towards giving professionals a wider toolkit to support treatment of all the different forms of PUP.

Ideally the next ten years will see a rising quality of research into PUP, moving us towards a landscape with more double-blind placebo trials for drugs and longitudinal studies covering time periods that allow us to take a broader view on PUP. It is a disorder that is slow to develop and potentially also slow to heal.

We should encourage better and deeper validation of a few assessment tools across diverse populations. Standardisation is helpful and surely two proven ones are sufficient. Link this to a strong communication programme to make them known widely in communities from general practitioners (MDs) and urologists to sex therapists, relationship counsellors, school counsellors and parents.

The next decade should see the debate around pornography use and erectile health fully resolved. Dialogues about the impact of pornography use in the context of violence against women and children also require more experimental approaches if the discussion is going to be able to move from correlation to causation.

The next few years will clearly be a time of great advances in linking genetic data and behaviour and we will monitor the contributions that this form of high-technology can make to epigenetic changes in humans.

### 5.3. Research for Consumers

At the low-tech end, the most important thing is to initiate a research and development programme to determine if early exposure to Internet pornography can be recognised as an adverse childhood experience (ACE). Success in that project could have a profound impact on raising awareness of the need to reduce early exposure and thereby reduce the risk of subsequent PUP.

Work monitoring the success, or otherwise, of age verification legislation around the world will be important for policy development. For individuals and communities, it is important that we learn which interventions are sustainable and cost effective.

Could commercial pornography suppliers be encouraged to take a pro-active role in heading off the development of PUP, rather than following their current laissez faire view that a customer can never have too much pornography? It will be important to have research to boost the accountability of suppliers. Accountability can be achieved through government intervention or industry self-regulation. So far the latter does not seem to be working in the consumer’s favour. In general the access to pornography is free, but the consumption risks are all transferred to the user.

The powerful position of commercial pornography suppliers within the context of the attention economy leaves the average consumer poorly informed about how their tastes and behaviour are being influenced by ‘free’ pornography. Commercial pornography suppliers use the data that gather to develop a deep understanding of consumer behaviour. Commercial players also benefit from the access to artificial intelligence and other data-mining tools to build market share and develop new markets.

### 5.4. Research for the Recovery Community

Good academic studies of the recovery communities, be they NoFap, Sex Addicts Anonymous or others, could be revolutionary. Recovery communities are a cheap solution to a potentially very expensive medical problem. However, often the communities do not really want to be studied. Members just want to be healed or to help others to heal. There is some acceptance that good research could help. There is also a recognition that for some PUP members, recovery communities are only part of the answer and that they may also need help from therapists and other professionals.

Quantitative research looking at the elimination of pornography is the highest priority. Does it really reverse sexual dysfunctions and have a positive impact on a variety of mental health disorders? The field currently has extensive anecdotal evidence to support these suggestions, but that is not a substitute for large-scale, well-grounded studies.

### 5.5. Moving Forward

It is now nearly two years since the Manifesto was published. In that time there has been a continuing growth world-wide in the volume of research appearing on pornography use in general and PUP in particular. However, most of the work being done tends to be at the easier, cheaper end of the scale. As far as we can tell as frequent observers of this field, there remains a general lack of vision and large-scale co-ordination between the many research groups around the world. The future needs to be more about designing investigations to test causation, not just to demonstrate yet more correlations.

The time is right for the COST Action team and the EU-PUI Network to create, fund and promote a visible and trusted champion for PUP research. Europe has sufficient resources to support such a role. Perhaps the research ideas sketched in this paper could be a positive contribution towards COST Action accepting it should deliver a leadership role in the field of PUP.

### 5.6. Limitations

Focusing on the Manifesto is a limiting strategy, in that many researchers in the pornography field around the world are unlikely to be familiar with it. Our approach has been to work with what is already available to build a more coherent research environment. At the same time, focusing on the Manifesto is essential. It is the only major published document which deals with the whole policy context of problematic use of the Internet. It is Europe-wide in its reach and is designed to endure for a decade.

By writing a response homing in on the issues around PUP, we hope to encourage researchers to have conversations to advance the field in ways that will benefit the continent’s taxpayers and publicly funded research programmes.

Our methodology of simply analysing all of the Manifesto as a whole through the particular lens of PUP is a limitation. This approach was taken to balance achieving the clearest level of understanding of what readers could infer from the document with an open agenda to build the strongest possible set of recommendations to shape the future of PUP research in Europe over the next decade.

## Figures and Tables

**Table 1 ijerph-17-03462-t001:** Summary of key research priorities to advance the understanding of PUI [1]

*1.* *“Reliable consensus-driven conceptualisation of PUI (defining main phenotypes and specifiers, related comorbidities and brain-based mechanisms)* *2.* *Age- and culture-appropriate assessment instruments to screen, diagnose and measure the severity of different forms of PUI* *3.* *Characterise the impacts of different forms of PUI on health and quality of life* *4.* *Define the clinical courses of different forms of PUI* *5.* *Reduce obstacles to timely recognition and interventions* *6.* *Clarify the possible role of genetics and personality features in different forms of PUI* *7.* *Consider the impact of social factors in the development of PUI* *8.* *Generate and validate effective interventions, both to prevent PUI, and to treat its various forms once established* *9.* *Identify biomarkers, including digital markers, to improve early detection and intervention”.*

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
