# Peer review of "Aligning the “Manifesto for a European Research Network into Problematic Usage of the Internet” with the Diverse Needs of the Professional and Consumer Communities Affected by Problematic Usage of Pornography"

_ijerph, 2020, doi:10.3390/ijerph17103462_

Round 1
Reviewer 1 Report
This manuscript reports the results of a text analysis of the “Manifesto for a European research network into Problematic Usage of the Internet” with a goal of aligning the current needs of clinicians and others affected by problematic pornography use. This is an interesting area. I offer the following suggestions which may improve the paper.
Introduction: The authors provide clear background on the “Manifesto” including the goal of developing policy related to research in behavioral addictions. There are no clear research questions and no clear rationale for the current paper.
Methods: The majority of the methods section consists of copies of sections of the “manifesto.” What is added by the authors of the current paper is primarily subject headings that guide the reader through the flow of the Manifesto. The authors add how they learned of the Manifesto and indicated how little the pornography as problematic usage of the internet was addressed. This paper was described in the abstract as a “textural [sic] analysis.” With a text analysis, the methods section should include a mention of the methods used to determine which passages to include and whether the authors collaborated on how to come to some consensus on what to include and how it aligns with the results later reported. There is no information on the methods or processes used in analyzing the original document.
Results: The results section reads like a literature review and does not incorporate information from an analysis of the text that the authors say is the framework for the paper (the Manifesto). It would have been good in this section to list the identified research areas from the manifesto and how the framework has been helpful or not helpful in advancing the research into problematic use of pornography.
Discussion: In the discussion section, the authors do refer back to the primary research areas identified in the Manifesto. Here the authors discuss the state of the research since the Manifesto was published and offer directions for future research. Although there is some important information presented here, it often does not align with the identified research areas. The authors state they are using the Manifesto as a framework but often veer from that path to explore other areas without explaining the connection.
For example, under Research area 3: Impacts of PUP on health and quality of life, the authors start with a discussion of whether PUP is a primary disorder. This part of the discussion provides relevant information on co-occurring mental health issues, personality traits, and other co-morbidities. The section on individual, relationship and community impacts has not been adequately tied back to the research area. Instead the authors discuss accessibility of pornography which seems tangential. The paragraphs on IPV may be relevant to health and quality of life but the authors do not tie it to those topics. The paragraph on tolerance and escalation might be better placed in the section on screening and assessment (Research area 2). The paragraphs on ASD do not fit under health and quality of life either as written.
Research area 4 according to the manifesto aims to define the clinical course of different forms of PUP. Instead the authors then focus on gender, lifestyle and sexuality of the individuals without discussing different forms of PUP. The closest the authors come to addressing this research area is in the discussion of PIED.
Discussions of research areas 5 and 6 were more true to the intended focus.
The conclusion mentions some key areas to address for future research. It reads like a list with several brief paragraphs that roughly align with each of the research areas. The authors do not include a recognition of limitations of the current research as reported. This might include the reliance on one main source which the authors admit does not sufficiently focus on the topic at hand followed by a clear statement on the lack of literature focusing on this topic. This would be the rationale for the paper. Limitations might also include the lack of a systematic method of analyzing the text and the literature reviewed in the paper as this is a possible source of bias. The conclusion could tie what was recommended in the original Manifesto to what was found in the current paper if the authors provide a clearer synthesis of the information included in the report, a recognition of gaps i nthe previous research, areas of overlap in the current paper, a call for action, and a clear plan for moving forward.
Author Response
Point 1: Introduction: The authors provide clear background on the “Manifesto” including the goal of developing policy related to research in behavioral addictions. There are no clear research questions and no clear rationale for the current paper.
Response 1: The research question has been added at lines 88 to 90 and more explanation for the rationale of the paper has been included in the introduction.
Point 2a: Methods: The majority of the methods section consists of copies of sections of the “manifesto.” What is added by the authors of the current paper is primarily subject headings that guide the reader through the flow of the Manifesto. The authors add how they learned of the Manifesto and indicated how little the pornography as problematic usage of the internet was addressed.
Response 2a: This point correctly describes what appears in our paper. it does not seem to need a response. The item about how we learned of the Manifesto has now been moved to the introduction in response to another reviewer's comments.
Point 2b: This paper was described in the abstract as a “textural [sic] analysis.” With a text analysis, the methods section should include a mention of the methods used to determine which passages to include and whether the authors collaborated on how to come to some consensus on what to include and how it aligns with the results later reported. There is no information on the methods or processes used in analyzing the original document.
Response 2b: We have changed the abstract to read "text " as recommended. We have stated that we have included 100% of the mentions of pornography in the Manifesto, so the recommendations about selecting parts are not relevant. By definition, our results fully align with what has been selected for discussion. We systematically view all 9 research areas of the Manifest through a PUP lens.
Point 3: Results: The results section reads like a literature review and does not incorporate information from an analysis of the text that the authors say is the framework for the paper (the Manifesto). It would have been good in this section to list the identified research areas from the manifesto and how the framework has been helpful or not helpful in advancing the research into problematic use of pornography.
Response 3: We think that there is some merit in this point. The reviewer wants the material we provide to answer this in the 'Results', whereas we have placed it in the 'Discussion'. We recommend that the contents of the 'Discussion' remain where they are at present.
In 'Results' we do deliver on the way the research areas of the Manifesto are helpful in advancing research for each of the three target groups - professionals, recovery groups and pornography consumer communities - covered by the research question.
Point 4: Discussion: In the discussion section, the authors do refer back to the primary research areas identified in the Manifesto. Here the authors discuss the state of the research since the Manifesto was published and offer directions for future research. Although there is some important information presented here, it often does not align with the identified research areas. The authors state they are using the Manifesto as a framework but often veer from that path to explore other areas without explaining the connection.
Response 4: We have aligned the discussion more closely to the Manifesto. This point is addressed by points 6 to 9 below.
Point 5: For example, under Research area 3: Impacts of PUP on health and quality of life, the authors start with a discussion of whether PUP is a primary disorder. This part of the discussion provides relevant information on co-occurring mental health issues, personality traits, and other co-morbidities.
Response 5: This point does not seem to require a response. It says that this part of the manuscript is satisfactory and is doing what it should.
Point 6: The section on individual, relationship and community impacts has not been adequately tied back to the research area. Instead the authors discuss accessibility of pornography which seems tangential.
Response 6: We have tied the issues of dyadic and individual health and wellbeing more closely to the existing literature, recommending research based on current gaps in the literature.
Point 7: The paragraphs on IPV may be relevant to health and quality of life but the authors do not tie it to those topics.
Response 7: It has now been tied together, lines 362 to 371.
Point 8: The paragraph on tolerance and escalation might be better placed in the section on screening and assessment (Research area 2).
Response 8: Agreed. It has been moved to Research area 2.
Point 9: The paragraphs on ASD do not fit under health and quality of life either as written.
Response 9: Accepted. We have added a longer introduction and moved the material on ASD to the end of Research area 2.
Point 10: Research area 4 according to the manifesto aims to define the clinical course of different forms of PUP. Instead the authors then focus on gender, lifestyle and sexuality of the individuals without discussing different forms of PUP. The closest the authors come to addressing this research area is in the discussion of PIED.
Response 10: We agree that the alignment in this section was poor and we have undertaken an extensive re-write of Research area 4, with a much larger focus on the 'clinical courses of different forms of PUP'
Point 11: Discussions of research areas 5 and 6 were more true to the intended focus.
Response 11: This is agreeing with the paper and no response is required.
Point 12: The conclusion mentions some key areas to address for future research. It reads like a list with several brief paragraphs that roughly align with each of the research areas. The authors do not include a recognition of limitations of the current research as reported. This might include the reliance on one main source which the authors admit does not sufficiently focus on the topic at hand followed by a clear statement on the lack of literature focusing on this topic. This would be the rationale for the paper. Limitations might also include the lack of a systematic method of analyzing the text and the literature reviewed in the paper as this is a possible source of bias. The conclusion could tie what was recommended in the original Manifesto to what was found in the current paper if the authors provide a clearer synthesis of the information included in the report, a recognition of gaps in the previous research, areas of overlap in the current paper, a call for action, and a clear plan for moving forward.
Response 12: We have now divided the conclusion into 6 sections. The first 4 allow a more specific examination of what research would benefit which target group, as per the research question. They tie the conclusions to the rationale for doing the study. We end by including a call to action and acknowledging the limitations coming from analysing a single, general policy document.
Reviewer 2 Report
The manuscript covers a relevant topic with an original point of view, and it could be of interest for IJERPH’s readers. The main reference works have been considered and the paper include a balanced, comprehensive and critical view of the research area. Method and data analysis are correct and well explained; results, summaries and bibliography are given appropriately in compliance with editor standards. However, in order to improve the quality of the manuscript, I suggest to include the following changes:
Sometimes in the text you talk about acronyms that are not specified for readers (for example, ISPs, at row 474, pag. 11). Please always specify them.
All the sub-chapter “Demand-site preventative interventions” (pag. 11, rows 481-496) should be rewritten. It does not include references and do not take care of the possible benefits of pornopgraphy. I know it is difficult to distinguish a useful use of porn than a problematic one, but I think that this paragraph could be mis-interpreted by some readers.
One of the research needs that has not been addressed in the article is the importance of focusing on the sexual preferences (quality, quantity and fixity) and on the excitment level felt for it, since they were prove to influences the risk of addiction (you can refer to Tripodi et al., 2015):
Tripodi F., Eleuteri S., Giuliani M., Rossi R., Livi S., Petruccelli I., Petruccelli F., Daneback K., Simonelli C. 2015, Unusual online sexual interests in heterosexual Swedish and Italian university students, Sexologies, 24, e84-e93.
Author Response
Point 1: The manuscript covers a relevant topic with an original point of view, and it could be of interest for IJERPH’s readers. The main reference works have been considered and the paper include a balanced, comprehensive and critical view of the research area. Method and data analysis are correct and well explained; results, summaries and bibliography are given appropriately in compliance with editor standards. However, in order to improve the quality of the manuscript, I suggest to include the following changes:
Response 1: This says we have done things right and there is no need to make adjustments based on this point.
Point 2: Sometimes in the text you talk about acronyms that are not specified for readers (for example, ISPs, at row 474, pag. 11). Please always specify them.
Response 2: Agreed and changed. ISP expanded to Internet Service Provider. No other unexpanded acronyms were identified.
Point 3: All the sub-chapter “Demand-site preventative interventions” (pag. 11, rows 481-496) should be rewritten. It does not include references and do not take care of the possible benefits of pornography. I know it is difficult to distinguish a useful use of porn than a problematic one, but I think that this paragraph could be mis-interpreted by some readers.
Response 3: We accept that there are not citations to literature in this section and have added some relevant ones. However, this absence is mainly because the formal research has not been done as yet. Rather than rewrite this paragraph, we have changed it a little for clarity.
To address the reviewer's concerns about the benefits of porn use, we have added a long paragraph placed immediately before it. This explores the issue the potential for users to develop PUP and considers benefits, risks and factors such as the time span of use..
Point 4: One of the research needs that has not been addressed in the article is the importance of focusing on the sexual preferences (quality, quantity and fixity) and on the excitment level felt for it, since they were prove to influences the risk of addiction (you can refer to Tripodi et al., 2015):
Tripodi F., Eleuteri S., Giuliani M., Rossi R., Livi S., Petruccelli I., Petruccelli F., Daneback K., Simonelli C. 2015, Unusual online sexual interests in heterosexual Swedish and Italian university students, Sexologies, 24, e84-e93.
Response 4: Accepted. Text added to lines 444 to 449 and citation added as [62] Warning. My copy of Zotero was playing up, so i only added this citation manually.
Reviewer 3 Report
overview
This is very timely in that it critically reviewed the proposed direction in the Manifesto regarding the PUP of research from the perspective of the actual experienced and presented a more specific direction.
Materials and Methods
Since it is a qualitative study and a proposed study of policy issues, it is limited to mention improvements in methodology.
However, if the authors want to comment on the less mention of pornography on manifesto, more accurate grounds are needed
That is, other sub-regions of Internet addiction in different areas such as gaming, gambling and streaming or social networks should also be examined for what percentage they are being mentioned. Furthermore, it is also necessary to take into account that Manifesto refers mainly to the entire Internet, rather than to addiction in any particular field, when describing the research priorities of each research area.
So, I would like to recommend that the authors change the paragraphs(line 134 ~ 139) based on the above comments
Results
Professional communities affected by problematic usage of pornography(line 159 ~ 176)
In the area of addiction, therapists' knowledge, attitudes and prejudices are very important issues in terms of access to appropriate treatment services. In particular, pornography addiction is an area that should be fully considered in terms of cross-cultural differences, as it becomes more taboo in Eastern culture. These aspects should be considered and mentioned in this section.
Consumer communities affected by problematic usage of pornography(line 219~240)
In this part, research on the effectiveness of voluntary consumer movements and activities are need to be considered importantly. Some more initiatives by NGO like “Common Sensehttps://www.commonsense.org/)could be mentioned in this section.
Discussion
Research area 3
As well as the issue of viewing child abuse materials, acts such as illegal filming and illegal sharing are treated as sexual crimes, making it an issue worldwide. I recommend that the authors could comment about this issue in lines (309 ~ 315)
Research 8
Supply-side prevention interventions
Accessibility limits has been established in other areas of addiction. This approach can be understood through public health model or framework including the tool from legal enforcement to voluntary movement by industry or monitoring system from civil movement by related NGOs.
The need for research on the frame of this environmental and institutional approach and its effectiveness needs to be mentioned in this section(line 471~475)
Conclusion
The authors mentioned about the role of commercial pornography suppliers. However, if we think about internet ecosystem, the internet or social network platform industries also should participate in the cooperative work. So, the authors should mention about the role of these parts(internet platform) as well as pornography supplier.
Author Response
Point 1: overview
This is very timely in that it critically reviewed the proposed direction in the Manifesto regarding the PUP of research from the perspective of the actual experienced and presented a more specific direction.
Response 1: This doe not require a response.
Point 2: Materials and Methods
Since it is a qualitative study and a proposed study of policy issues, it is limited to mention improvements in methodology.
However, if the authors want to comment on the less mention of pornography on manifesto, more accurate grounds are needed
That is, other sub-regions of Internet addiction in different areas such as gaming, gambling and streaming or social networks should also be examined for what percentage they are being mentioned. Furthermore, it is also necessary to take into account that Manifesto refers mainly to the entire Internet, rather than to addiction in any particular field, when describing the research priorities of each research area.
So, I would like to recommend that the authors change the paragraphs(line 134 ~ 139) based on the above comments
Response 2: There may be value in looking in more detail at the other forms of PUI, and the extent they are also considered throughout the Manifesto, but we feel that to do so would be outside the scope of our study, as doing so would only marginally impact on what research needs to be done to provide a better understanding of PUP. This paper is not about how well (or not) other areas of problematic use of the internet have been treated by the Manifesto. It is about what research should be done on PUP to meet the needs of specific audiences. Research is more likely to be commissioned, funded or supported if it fits into the policy funding framework. That means being mentioned in the appropriate parts of the Manifesto.
Point 3: Results
Professional communities affected by problematic usage of pornography(line 159 ~ 176)
In the area of addiction, therapists' knowledge, attitudes and prejudices are very important issues in terms of access to appropriate treatment services. In particular, pornography addiction is an area that should be fully considered in terms of cross-cultural differences, as it becomes more taboo in Eastern culture. These aspects should be considered and mentioned in this section.
Response 3: This section has been changed and now begins at line 182. We have incorporated the ideas of this point in our study at lines 195 to 201.
Point 4: Consumer communities affected by problematic usage of pornography(line 219~240)
In this part, research on the effectiveness of voluntary consumer movements and activities are need to be considered importantly. Some more initiatives by NGO like “Common Sensehttps://www.commonsense.org/)could be mentioned in this section.
Response 4: This section now starts at line 249. We have looked carefully at https://www.commonsense.org and it seems to be irrelevant to the prevention of PUP. It has one resource on discouraging sexting in 8th grade students. We are aware of better resources. Instead, in response to this suggestion, we have rewritten lines 257 to 263 and added reference to the research needs of three other larger-scale NGOs who are major providers in the PUP education space.
Point 5: Discussion
Research area 3
As well as the issue of viewing child abuse materials, acts such as illegal filming and illegal sharing are treated as sexual crimes, making it an issue worldwide. I recommend that the authors could comment about this issue in lines (309 ~ 315)
Response 5: This section has now been moved forward to line 298 to 308. We accept this suggestion, but rather than placing it where they have suggested, we have added it to lines 402 to 404.
Point 6: Research 8
Supply-side prevention interventions
Accessibility limits has been established in other areas of addiction. This approach can be understood through public health model or framework including the tool from legal enforcement to voluntary movement by industry or monitoring system from civil movement by related NGOs.
The need for research on the frame of this environmental and institutional approach and its effectiveness needs to be mentioned in this section(line 471~475)
Response 6: This is a useful idea and we have incorporated new material based on this at lines 551 to 556.
Point 7 Conclusion
The authors mentioned about the role of commercial pornography suppliers. However, if we think about internet ecosystem, the internet or social network platform industries also should participate in the cooperative work. So, the authors should mention about the role of these parts(internet platform) as well as pornography supplier.
Response 7: Agreed. We have added material at lines 557 to 561.
Reviewer 4 Report
Aligning the “Manifesto for a European research network into Problematic Usage of the Internet” with the Diverse Needs of the professional and consumer communities affected by problematic usage of pornography
Reviewer’s Comments
I wonder if this study should be considered as original article.
It seems very confusing because the structure of the content written in the Introduction-Method-Results is not familiar. In the introduction, it is necessary to present the opinions on the issues to be addressed and the significance to be dealt with for Manifesto, which is the background of the research, and to present the methods and results. The text analysis of Manifesto in the research method reveals its impact and diverse needs. I think it should be discussed based on the research results.
Line 52 It needs more supplementary explanation to support the rationale. The impact of professions, the recovery communities, and consumer needs further explanation about pornography usage and its relevance to PUI.
Line 58 This study is a methodological study using text analysis. Therefore, it is necessary to describe in detail how two researchers analyzed the text analysis method and the subject of study (Manifesto).
Line 134, Line 139, etc. seems like it is background of the study, so I wonder if it should be in the introduction part or not...
Line 149 It was mentioned in the introduction, Shouldn't the study results be described according to the purpose of the research? As mentioned earlier, it seems necessary to revise the introduction-method-result.
Author Response
Point 1: I wonder if this study should be considered as original article.
Response 1: This is a decision for IJERPH, not the authors. We are happy to be published under whatever title and form MDPI recommend as publishers.
Point 2: It seems very confusing because the structure of the content written in the Introduction-Method-Results is not familiar. In the introduction, it is necessary to present the opinions on the issues to be addressed and the significance to be dealt with for Manifesto, which is the background of the research, and to present the methods and results. The text analysis of Manifesto in the research method reveals its impact and diverse needs. I think it should be discussed based on the research results.
Response 2: The structure is imposed by IJERPH. We were not very comfortable with it, but have gone to a lot of effort to comply with the standard set structure. We think that adding the research question to point 3 helps.
Point 3: Line 52 It needs more supplementary explanation to support the rationale. The impact of professions, the recovery communities, and consumer needs further explanation about pornography usage and its relevance to PUI.
Response 3: We have added a clear research question at lines 88 to 90.
Point 4: Line 58 This study is a methodological study using text analysis. Therefore, it is necessary to describe in detail how two researchers analyzed the text analysis method and the subject of study (Manifesto).
Response 4: We have added clarification to the first paragraph in Materials and methods, line 92 onwards. At the same time, we see our paper as an analysis of policy and not as a contribution to knowledge in the field of text analysis, so our focus is on the substance of identifying the research areas needed to support policy, not on improving academic text analysis techniques.
Point 5: Line 134, Line 139, etc. seems like it is background of the study, so I wonder if it should be in the introduction part or not...
Response 5: We agree that this could be moved to the introduction, but we do not propose to do so. It is a significant research finding - the fact that there is only one mention of pornography after Section 1 of the Manifesto is significant, as no non-specialist reader is likely to pick up the fact that pornography has not been named, while other PUIs such as gaming, gambling etc have been named. As a research finding, it is currently located in the correct place.
Point 6: Line 149 It was mentioned in the introduction, Shouldn't the study results be described according to the purpose of the research? As mentioned earlier, it seems necessary to revise the introduction-method-result.
Response 6: We have moved the discussion of the size of the pornography market to the introduction and moved citations [3-6] to line 59. The study results now align clearly with the research question set out in lines 88 to 90
Reviewer 5 Report
Overall this is an unusual paper, which has an accptable academic structure of introduction, methods, results, and discussion. However, this does not present data per se but rather reports on a manifesto and offers detals as to one specific aspect of problematic ussage of the internet, i.e. pornographic usage, should be identifed, conceptualized, and studied. The research areas of interest are appropriate and well described. Overall, this adds to the literature that is evolving concerning behavioral addictions and that involving specifically compulsive sexual behavior disorder and, as such, is a good contribution.
Author Response
Point 1: Overall this is an unusual paper, which has an acceptable academic structure of introduction, methods, results, and discussion. However, this does not present data per se but rather reports on a manifesto and offers details as to one specific aspect of problematic usage of the internet, i.e. pornographic usage, should be identified, conceptualized, and studied. The research areas of interest are appropriate and well described. Overall, this adds to the literature that is evolving concerning behavioral addictions and that involving specifically compulsive sexual behavior disorder and, as such, is a good contribution.
Response 1: Positive comments, no action required.
Round 2
Reviewer 1 Report
This is a resubmission of a manuscript that reports the results of a text analysis of the “Manifesto for a European research network into Problematic Usage of the Internet” with a goal of aligning the current needs of clinicians and others affected by problematic pornography use. Although the paper is improved, there are still some areas needing attention.
Introduction:
The addition of the section on problematic usage of pornography in the context of PUI is an improvement and helps to support the rationale for your study. Paragraphs 2-4 have no citations. Please cite sources of the information you provided in those paragraphs. Below are a couple of reviews I have found on pornography addiction and treatment and one study of an evidence-based self-help group intervention. You mention consumer discomfort with the impacts of pornography use seeking treatment with no research evidence to support your statements. I am not recommended any of these articles but some references are definitely needed here.
Doring, N. M. (2009). The Internet’s impact on sexuality: A critical review of 15 years of research. Computers in Human Behavior, 25, 1089–1101.
Duffy, A., Dawson, D. L., & Das Nair, R. (2016). Pornography addiction in adults: A systematic review of definitions and reported impact. The Journal of Sexual Medicine, 13, 760–777.
Levin, Michael E et al. (2017). Examining the feasibility of Acceptance and Commitment Therapy self-help for problematic pornography viewing: Results from a pilot open trial. The Family Journal 25(4), 306–312.
Line 73 (edited version) should correct Manifest to Manifesto.
When you state “We base our work on people with lived-experience and on making the insights from the formal and informal research accessible to a wider readership,” are you referring to “our work” on this particular paper or more broadly? I do not think detailing your meeting with Burkhausas adds to this paper at all, nor does identifying the authors’ affiliation as stakeholders. I suggest including Burkhausas in an acknowledgements statement. The paragraph starts with the statement that “There is evidence that the COST Action Dissemination Plan is working” with this meeting as evidence of that. I am not convinced the paragraph is needed at all. If you include it, it can probably be simplified to one sentence as an introduction to the next paragraph. “This paper is a result of the authors heeding this call to action.”
Methods:
The text in the methods section was reorganized. However, the methods section still lacks information on methods used in the text analysis as mentioned by multiple reviewers.
Reviewer 1 stated “With a text analysis, the methods section should include a mention of the methods used to determine which passages to include and whether the authors collaborated on how to come to some consensus on what to include and how it aligns with the results later reported. There is no information on the methods or processes used in analyzing the original document.”
Reviewer 3 stated “Since it is a qualitative study and a proposed study of policy issues, it is limited to mention improvements in methodology. However, if the authors want to comment on the less mention of pornography on manifesto, more accurate grounds are needed”
Reviewer 4 stated “This study is a methodological study using text analysis. Therefore, it is necessary to describe in detail how two researchers analyzed the text analysis method and the subject of study (Manifesto).”
None of these recommendations were incorporated into the methods section of the paper and this was not addressed as a limitation of the paper.
Line 182
“This section starts to indicate at least one area of potential future research. Is viewing pornography a PUI that meets the physiological criteria for addiction?”
In identifying this potential area of research, you might consider changes in DSM 5 for substance use disorders and gambling disorder where the typical physiological criteria previously applied to addiction (withdrawal and dependence) are not necessary to define a use disorder. Although pornography has not been addressed in DSM5 due to the lack of research to support diagnostic criteria, they did include gambling disorder where the physiological criterion is irritability and restlessness when trying to cut down or stop. Similarly Internet Gaming Disorder is included in the Conditions for Future Study section with withdrawal symptoms listed as irritability, anger, or sadness with no physical signs of pharmacological withdrawal.
Results: The results section is improved but some of it is still more like a literature review especially in places where information from the analysis of the text is not incorporated and the stated purpose of the paper is temporarily lost.
For example, information under the subsection of professional communities affected by PUP should be tied back to the purpose and the text analysis. Information in this section could be tied to conceptualization and assessment (1 and 2), obstacles (5), social factors (7), etc. However, since this is the results section of a text analysis, this should be explicitly stated rather than have the reader infer. Additionally, a paragraph was added about cultural taboos and blindspots in response to a suggestion from one of the reviewers. This is important but it hangs there with no real context or connection to the paragraph before and after it. The authors state “it is clear from the literature” and do not include any citations of sources.
I give feedback on this particular subsection but my comments apply to other subsections in the results section as well. It should all show how your text analysis of the source is connected to these results.
Discussion:
The discussion section is improved and better organized. Still be mindful of leaving paragraphs “orphaned.” For example, the paragraph comprised of lines 385-389 is not tied to the topic of screening, assessment, diagnosis. A sentence is needed to tie this to the other information in the section. Conclusion:
The conclusion mentions some key areas to address for future research. This is improved from the previous version. The authors did a good job of adding future directions and some of limitations. The authors acknowledge the limitation of having a single document as the primary source. The authors did not address the limitation of not systematically analyzing the text despite not including procedures in the methods section for limiting bias.
Author Response
Point 1: This is a resubmission of a manuscript that reports the results of a text analysis of the “Manifesto for a European research network into Problematic Usage of the Internet” with a goal of aligning the current needs of clinicians and others affected by problematic pornography use. Although the paper is improved, there are still some areas needing attention.
Response 1: We address each of the concrete suggestions in the following points.
Point 2: Introduction
The addition of the section on problematic usage of pornography in the context of PUI is an improvement and helps to support the rationale for your study. Paragraphs 2-4 have no citations. Please cite sources of the information you provided in those paragraphs. Below are a couple of reviews I have found on pornography addiction and treatment and one study of an evidence-based self-help group intervention. You mention consumer discomfort with the impacts of pornography use seeking treatment with no research evidence to support your statements. I am not recommended any of these articles but some references are definitely needed here.
Doring, N. M. (2009). The Internet’s impact on sexuality: A critical review of 15 years of research. Computers in Human Behavior, 25, 1089–1101.
Duffy, A., Dawson, D. L., & Das Nair, R. (2016). Pornography addiction in adults: A systematic review of definitions and reported impact. The Journal of Sexual Medicine, 13, 760–777.
Levin, Michael E et al. (2017). Examining the feasibility of Acceptance and Commitment Therapy self-help for problematic pornography viewing: Results from a pilot open trial. The Family Journal, 25(4), 306–312.
Response 2: One paragraph referred to does actually have evidence cited [3-6], but we concede that we have not referred to the academic literature here. We have added a total of 13 citations, 7 of which are new to this paper [63-69]. The suggestion of Doring [2009] by the reviewer was taken up, but the other two suggested papers are not really appropriate here.
Point 3: Line 73 (edited version) should correct Manifest to Manifesto.
Response 3: Agreed and done.
Point 4: When you state “We base our work on people with lived-experience and on making the insights from the formal and informal research accessible to a wider readership,” are you referring to “our work” on this particular paper or more broadly?
Response 4: Our meaning is “more broadly”. We have rewritten lines 77 to 80 to provide clarity. This should remove the ambiguity noted by Reviewer 1.
Point 5: I do not think detailing your meeting with Burkhausas adds to this paper at all, nor does identifying the authors’ affiliation as stakeholders. I suggest including Burkhausas in an acknowledgements statement. The paragraph starts with the statement that “There is evidence that the COST Action Dissemination Plan is working” with this meeting as evidence of that. I am not convinced the paragraph is needed at all. If you include it, it can probably be simplified to one sentence as an introduction to the next paragraph. “This paper is a result of the authors heeding this call to action.”
Response 5: We accept this suggestion, removing the material referring to Burkhausas from lines 77+ and substituting an Acknowledgement just before the References section at the end of the paper. The authors affiliations were also removed and the recommended text was used “This paper is a result of the authors heeding this call to action.”
Point 6: Methods:
The text in the methods section was reorganized. However, the methods section still lacks information on methods used in the text analysis as mentioned by multiple reviewers.
Reviewer 1 stated “With a text analysis, the methods section should include a mention of the methods used to determine which passages to include and whether the authors collaborated on how to come to some consensus on what to include and how it aligns with the results later reported. There is no information on the methods or processes used in analyzing the original document.”
Reviewer 3 stated “Since it is a qualitative study and a proposed study of policy issues, it is limited to mention improvements in methodology. However, if the authors want to comment on the less mention of pornography on manifesto, more accurate grounds are needed”
Reviewer 4 stated “This study is a methodological study using text analysis. Therefore, it is necessary to describe in detail how two researchers analyzed the text analysis method and the subject of study (Manifesto).”
None of these recommendations were incorporated into the methods section of the paper and this was not addressed as a limitation of the paper.
Response 6: We recognise that the reviewers became quite animated around methodologies relating to text analysis.
As a response, first we have expanded the description of what we have done (lines 86 to 95). Second, we have separately removed the term ‘text’ from the manuscript, changing it simply to ‘analysis’. In this way we believe that we have moved away from all of the methodological baggage tied up in ‘text analysis’. And lastly, we have added a further paragraph to the ‘Limitations’ section in the ‘Conclusions’, lines 770 to 774, acknowledging this as a limitation of the study.
Point 7: Line 182
“This section starts to indicate at least one area of potential future research. Is viewing pornography a PUI that meets the physiological criteria for addiction?”
In identifying this potential area of research, you might consider changes in DSM 5 for substance use disorders and gambling disorder where the typical physiological criteria previously applied to addiction (withdrawal and dependence) are not necessary to define a use disorder. Although pornography has not been addressed in DSM5 due to the lack of research to support diagnostic criteria, they did include gambling disorder where the physiological criterion is irritability and restlessness when trying to cut down or stop. Similarly Internet Gaming Disorder is included in the Conditions for Future Study section with withdrawal symptoms listed as irritability, anger, or sadness with no physical signs of pharmacological withdrawal.
Response 7: We agree with some of this analysis, but not all of it.
We are fully aware of the stories of the passing of pornography and substance use disorders, as well as gambling and gaming disorders, through the preparation of the most recent DSM. We also understand the passage of these disorders through the WHO’s International Classification of Diseases. The Manifesto does refer to both.
We do not agree with the Reviewers assertion that “Although pornography has not been addressed in DSM-5 due to the lack of research to support diagnostic criteria…” Firstly, there is a powerful research base that suggests pornography was excluded from DSM-5 for internal political reasons within the American Psychiatric Association, not for a lack of research to support the diagnostic criteria. Additionally, the research needed to fill this ‘perceived’ gap has now been completed and published, so there is no need for us to map it out for future work.
Second, at the most basic level, the history of the DSM is an American-centric issue. It is substantially less important in Europe, or through the rest of the world, where ICD-10 and ICD-11 are the international standards. DSM uses ICD codes, not the other way around. In the paper we have referenced the process of placing PUP in the ICD context for the European Manifesto.
We could explore this territory if the editors insist, but it is not an area where our discussion is likely to improve on the existing literature.
On this basis we do not propose to make any further edits around lines 122-133.
We have discussed the disease classification issue in some detail in lines 412 to 422.
Point 8: Results: The results section is improved but some of it is still more like a literature review especially in places where information from the analysis of the text is not incorporated and the stated purpose of the paper is temporarily lost.
For example, information under the subsection of professional communities affected by PUP should be tied back to the purpose and the text analysis. Information in this section could be tied to conceptualization and assessment (1 and 2), obstacles (5), social factors (7), etc. However, since this is the results section of a text analysis, this should be explicitly stated rather than have the reader infer.
Response 8: To respond to this point required some major shifts within the paper. These happened at the end of section 2, in section 3 and in the first part of section 4. In simple terms, we have made the results clearer and shorter, then extended the Discussion to approach the research question from more perspectives.
We have expanded the results based on our analysis in Section 3 by adding additional text and by moving the last part of section 2, ‘Pornography and the Manifesto’ to become the last part of Section 3. We then moved most of Section 3 into the early part Section 4. Section 4 ‘Discussion” now falls in two parts – Community-focused issues’ at line 194 and ‘Future research seen through the lens of the nine priorities’ at line 303.
Point 9: Additionally, a paragraph was added about cultural taboos and blindspots in response to a suggestion from one of the reviewers. This is important but it hangs there with no real context or connection to the paragraph before and after it. The authors state “it is clear from the literature” and do not include any citations of sources.
Response 9: This is a legitimate issue. The collective literature does support what we wrote, but we know of no studies which encapsulate this in a simple and definitive way. To properly justify the statement would take a whole separate paper. Therefore, we have removed the statement ‘it is clear from the literature’ at line 196. Instead we have added two citations to our own reports of conference observations which do support the rephrased sentence now beginning ‘It is apparent from our experience in meeting researchers…’.
Point 10: I give feedback on this particular subsection but my comments apply to other subsections in the results section as well. It should all show how your text analysis of the source is connected to these results.
Response 10: Noted. Resolved in 8.
Point 11: Discussion:
The discussion section is improved and better organized. Still be mindful of leaving paragraphs “orphaned.” For example, the paragraph comprised of lines 385-389 is not tied to the topic of screening, assessment, diagnosis. A sentence is needed to tie this to the other information in the section.
Response 11: Agreed. We have added ‘from the perspectives of screening, assessment and diagnosis’ to line 416 to strengthen the link.
Point 12: Conclusion:
The conclusion mentions some key areas to address for future research. This is improved from the previous version. The authors did a good job of adding future directions and some of limitations. The authors acknowledge the limitation of having a single document as the primary source. The authors did not address the limitation of not systematically analyzing the text despite not including procedures in the methods section for limiting bias.
Response 12: Accepted. We have now addressed this as set out in our response to Point 6, above.
Reviewer 2 Report
The manuscript covers a relevant topic with an original point of view, and it could be of interest for IJERPH’s readers. The main reference works have been considered and the paper include a balanced, comprehensive and critical view of the research area. Method and data analysis are correct and well explained; results, summaries and bibliography are given appropriately in compliance with editor standards. This new version is much improved and I think it can be published in the present form.
Author Response
Point 1: The manuscript covers a relevant topic with an original point of view, and it could be of interest for IJERPH’s readers. The main reference works have been considered and the paper include a balanced, comprehensive and critical view of the research area. Method and data analysis are correct and well explained; results, summaries and bibliography are given appropriately in compliance with editor standards. This new version is much improved and I think it can be published in the present form.
Response 1: Positive comments, no action required.
Reviewer 4 Report
I still wonder if this study should be considered as original article.
Despite the efforts of the authors, the content written here is not clearly described.
Overall, it is required scientific evidence for research methods and results.
Author Response
Point 1: I still wonder if this study should be considered as original article.
Response 1: In offering this article to IJERPH, the authors’ focus has been on responding specifically to theme of the issue - Internet and Smartphone Use-Related Addiction Health Problems: Treatment, Education and Research. We believe that we have written an original article which looks at the opportunities and needs linking PUP research to consumer communities. With over half a million people using a single recovery website to seek relief from physical and mental health problems associated with PUP, this is an issue invoking large-scale political and community interest. As such, it is worthy of considered, systematic analysis in helping plan Europe’s future research programmes in the area. We are unaware of any other similar studies for planning research into PUP anywhere in the world.
Point 2: Despite the efforts of the authors, the content written here is not clearly described.
Response 2: Two other reviewers disagree. However, by making further adjustments to the points made by Reviewer 1, we believe that we have strengthened the clarity with which the content is described in a way that should meet the standards of Reviewer 4.
Point 3: Overall, it is required scientific evidence for research methods and results.
Response 3: We have extensively rewritten the methods and results sections.
The ultimate goal of this paper is to encourage funders and people commissioning research into PUP to consider the needs of communities having health issues linked to PUP. We believe that we have provided sufficient rationale to support this goal.